# The effect of different materials under the fencing piste on impact shock of the tibia during the fencing lunge on a concrete surface

Katharine Holmes[1], Jonathan Sinclair[2], Ashley Titan[3], Camilla Holland[4], Michael Grebla [5], Lindsay Bottoms[4]*

1 Icahn School of Medicine at Mount Sinai, New York, New York, United States of America, 2 School of Sport and Health Sciences, University of Central Lancashire, Preston, United Kingdom, 3 Stanford School of Medicine, Department of Surgery, Division of Plastic Surgery, Stanford, California, United States of America, 4 Centre for Research in Psychology and Sports, University of Hertfordshire, Hatfield, United Kingdom, 5 Mr Michael Grebla, Independent Researcher, New York, New York, United States of America

◉ These authors contributed equally to this work.
* l.bottoms@herts.ac.uk, @BottomsLindsayy

## Abstract

Fencing has gained global popularity, with athletes often competing on hard surfaces, especially at United States national tournaments in convention centre with concrete floors. These surfaces may contribute to overuse injuries from high-impact movements like the fencing lunge. This study aimed to investigate tibial accelerations, a measure of impact shock, experienced by fencers during lunges on various surface materials placed beneath standard aluminium fencing pistes. The aim was to identify a material that could reduce injury risk by mitigating impact shock. Sixty-nine fencers (35 female) performed five lunges on six different surfaces (A–F: A–E composed of various materials placed between the aluminium piste and the concrete floor; F was only the concrete floor), during the 2024 US Senior National Championships. A triaxial accelerometer mounted on the tibia was used to measure tibial accelerations as a marker of impact shock. The accelerometer was aligned to measure acceleration along the longitudinal axis of the tibia and set to record at 1000 Hz with a sensitivity range of ± 100 g. Data acquisition was carried out via a logging system (Biometrics DL1001, Gwent, UK), which was attached to the participant using a tightly fitted backpack. The peak positive axial tibial acceleration was extracted for each lunge and the average was calculated from three lunges after discarding the highest and lowest values from each surface. Statistical analysis revealed that Surface E (a non-absorbent vinyl loop material; 12.7 ± 7.6g), significantly reduced tibial accelerations compared to the standard concrete setup (Surface F; 13.6 ± 8.4g). These findings suggest that modifying competition surfaces by incorporating cushioning materials may help reduce the impact shock of the fencing lunge, potentially lowering the risk of overuse injuries, such as tendonitis and tenosynovitis, commonly reported by fencers. Future research should investigate optimal material properties, including thickness and

**Data availability statement:** All relevant data for this study are publicly available from the University of Hertfordshire Research Archive repository (https://doi.org/10.18745/ds.28796).

**Funding:** The author(s) received no specific funding for this work.

**Competing interests:** The authors have declared that no competing interests exist.

softness, for maximizing injury prevention while maintaining performance standards in competitive fencing environments.

## Introduction

Fencing is one of the only four sports (athletics, cycling, swimming and fencing) to be included in every iteration of the Modern Olympic Games since their inception in 1896 [1]. Around the world and within the United States, fencing has been growing in popularity, with USA Fencing alone counting almost 40,000 members as of 2024 [2]. Within the United States, there are between five to eight national competitions a year, with upwards of 5,000 registered athletes [2]. These competitions are all held in convention s across the country and the field of play is placed directly on concrete floors. The field of play consists of fencing pistes- grounded metal strips (often aluminum or copper) that are 14 meters long and 1.5 meters wide [3]. However, there is no requirement regarding piste thickness, which can range from thin mats of woven copper a few millimeters thick to planks of aluminum 2–3 cm thick. Fencing pistes have little to no padding, and fencers at US Fencing national tournaments must compete directly on metal laid on concrete.

Previous research has shown that the impact shock of the tibia varies greatly depending on the material an individual is lunging on. For example, lunging directly on concrete with an overlaid vinyl layer produces significantly greater impact shock of the tibia than lunging on a wooden sprung court surface or wooden sprung court surface overlaid with an aluminum piste [4]. This is concerning when considering past findings that the fencing lunge exposes participants to potentially detrimental impact shock and as such has been shown to place the fencer's musculoskeletal system under stress and increase risk of injury [5,6].

More generally, it is known that the transient shockwave generated by a heel striking the ground propagates through the musculoskeletal system, increasing injury risk [7]. Since the dominant leg absorbs most of the impact during a fencing lunge, it is more prone to injury [8,9].

Other studies have also suggested a direct correlation between the magnitude of impact shock, frequency of repetition, and the development of overuse injuries [10,11]. This aligns with prior research into in competition injuries that identified tendonitis, strains, and sprains, particularly of the knee and ankle as the most common injuries [8,9].

It has been suggested that improving surface cushioning to reduce the impact of movements can help lower the risk of injury [12–14]. This study aimed to explore whether placing different materials under metal fencing pistes could effectively reduce tibial accelerations during lunges and thereby mitigate injury risk. We conducted this study at the 2024 US Senior National Championships/April North American Cup, as laboratory-based material testing has produced inconsistent results when predicting the load on the musculoskeletal system during sports-specific movements [15]. Studies have shown that calculating the hardness of a surface and collecting the ground reaction forces experienced by athletes yield significantly

different forces [16,17]. To explore this, we had fencers lunge on six identical fencing pistes, with five pistes placed on various materials and one directly on concrete. We aimed to identify a material that could reduce tibial accelerations during lunges, ultimately reducing injury risk.

## Materials and methods

### Participants

Seventy fencers (35 female) volunteered to participate in the study. Fencers' characteristics can be seen in Table 1. Participants had at least 1 year experience of training in any fencing weapon (épée, foil and sabre) and were participating in the National American Championships for Fencing in April 2024 at the Salt Lake City Convention. Consent was obtained from USA Fencing to recruit and undertake data collection at the Championships prior to data collection. Participants completed health screens to determine that they were free from injury and provided written informed consent. Institutional ethical approval was obtained from 2 Universities (one UK and one USA based) for the study (protocol number: aLMS/SF/UH/ 05582 and 70791) was obtained in accordance with the principles outlined in the Declaration of Helsinki. Once consent was provided, participants were given an ID number for data collection and analyses. A priori power analysis was conducted to reduce the likelihood of a type II error and to determine the minimum number of participants needed for this investigation. It was found that the sample size was sufficient to provide more than 80% statistical power in the experimental measure between surfaces.

### Study design

This study used an observational design where the fencers completed all conditions. Each fencer performed 5 lunges on each of the 6 pistes with the different materials underneath or concrete only. The materials were provided by Action Floor Systems and were commercially available at the time of data collection, designed to absorb high impact and easily placed under the fencing pistes. The pistes were labelled A to F and the order was randomised for each fencer. Table 2 provides the properties of each of the surfaces.

### Procedures

A triaxial accelerometer (Biometrics S3-1000G-HA, Gwent, UK) was mounted on a lightweight carbon-fibre plate and affixed to the distal anteromedial region of the tibia, 8 cm proximal to the medial malleolus of the front leg. This location was selected based on prior studies [18] to facilitate comparison with earlier research examining impact shock during a fencing lunge [5]. The carbon plate was secured to the participant's shank using strong adhesive tape, applied as tightly as possible without causing significant discomfort (Fig 1). To ensure a rigid coupling between the accelerometer and the tibia, the underlying skin was stretched, enhancing the mounted device's resonance frequency to exceed 70 Hz. The accelerometer was aligned to measure acceleration along the longitudinal axis of the tibia, set to record at 1000 Hz with a

**Table 1. Fencers' characteristics: age, stature (H), body mass (BM) and body mass index (BMI).**

|  | Male (n = 34) | Female (n = 35) | All (n = 69) |
|---|---|---|---|
| Age (years) | 37.3 ± 18.8 | 48.2 ± 18.7 | 42.1 ± 19.6 |
| H (cm) | 181.9 ± 8.2 | 167.3 ± 7.9 | 174.4 ± 10.8 |
| BM (kg) | 80.9 ± 12.3 | 67.3 ± 9.8 | 74.0 ± 13.0 |
| BMI (kg·m⁻²) | 24.4 ± 2.6 | 24.1 ± 3.3 | 24.2 ± 3.0 |

Values are Mean ±SD

**Table 2. Properties of each of the surfaces used.**

| Sur-face | Description | Thickness (mm) | Density (kg.m⁻³) | Tensile Strength (psi) |
|---|---|---|---|---|
| A | Action 404 Rubber Underlayment (single layer) | 7 | 800 | 178 |
| B | Double layer of Surface A | 14 | 800 | |
| C | Herculan Cushion MF Blue rebound foam (single layer) | 7 | 310 | >50.7 |
| D | Double layer of Surface C | 14 | 310 | |
| E | Non-absorbent vinyl loop coils extruded from 100% PVC, thermally bonded (single layer) | 12.7 | 303.6 | 0.6 |
| F | Concrete floor (no surface material between the concrete floor and piste) | | | |

Fencers wore shoes that they fence in and were instructed to perform their own warmup. The experiment was carried out during the competition. All the lunges were completed in a convention hall on a concrete floor.

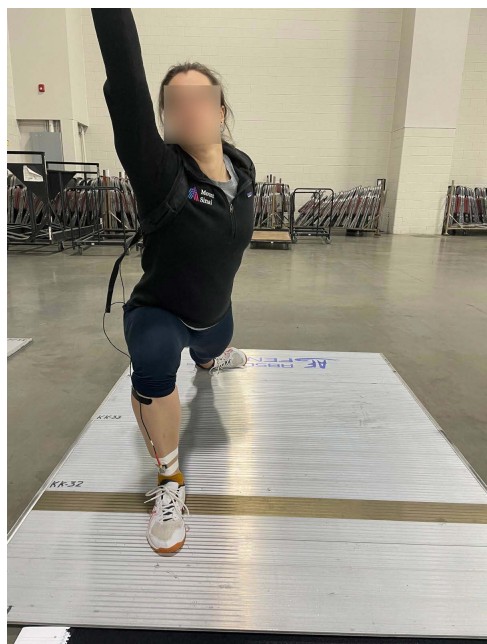

**Fig 1. Demonstration of the lunge on the piste with the accelerometer mounted on the front leg.**

sensitivity range of ± 100 g. Data acquisition was carried out via a logging system (Biometrics DL1001, Gwent, UK), which was attached to the participant using a tightly fitted backpack.

## Data processing

Tibial acceleration data were analysed using Biometrics DataLITE Management Software (Version 11.02) [19] for quantification and processing. Prior to conducting the data analysis, the acceleration signals underwent filtering using a 60 Hz Butterworth zero-lag, second-order low-pass filter [5]. This filtering process was applied to mitigate any potential resonance effects on the acceleration signal. The peak positive axial tibial acceleration was extracted for each lunge and an average was calculated from three after discarding the highest and lowest values from each surface.

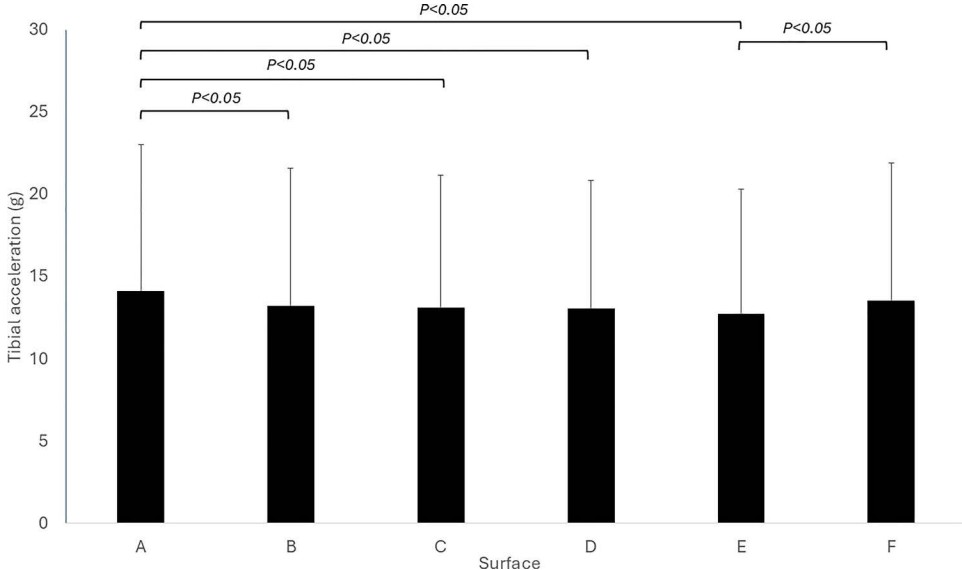

**Fig 2. Mean ±SD tibial accelerations for each surface.**

## Statistical analysis

Means and standard deviations (SD) were computed for each experimental surface. Statistical comparisons between surfaces were conducted using within-subjects linear mixed-effects models, employing compound symmetry and restricted maximum likelihood estimation techniques. In these models, participants were treated as random intercepts, while age (<40 and ≥40), sex (male and female), and weapon type (épée, saber, foil, and those participating in multiple disciplines) were included as covariates. In addition, the proportion of fencers that experienced both their greatest and lowest tibial accelerations on each surface were examined using one-way Chi-squared ($X^2$) goodness of fit tests. Statistical significance was set at P ≤ 0.05 for all analyses. All statistical analyses were performed using SPSS version 29 (IBM, SPSS, Armonk, NY, USA).

## Results

The results revealed that surface A had significantly greater tibial accelerations (14.1 ± 8.9g) than B (13.2 ± 8.3g; P = 0.037), C (13.1 ± 8.0g; P = 0.04), D (13.1 ± 7.8g; P = 0.014) and E (12.7 ± 7.6g; P = 0.007). Furthermore, the results also revealed that tibial accelerations were significantly greater in surface F (13.6 ± 8.4g) compared to surface E (12.7 ± 7.6g; P = 0.008) (Fig 2).

As can be seen in Table 3, for the proportion of fencers experiencing their greatest tibial accelerations on each surface, the chi-squared test showed that more fencers experienced their highest tibial acceleration on surface A ($X^2$ (5) = 14.74, P = 0.012). Similarly, for the number experiencing their lowest tibial accelerations on each surface, the chi-squared test showed that more fencers experienced their lowest tibial acceleration on surface E ($X^2$ (5) = 14.04, P = 0.015).

## Discussion

The purpose of this study was to find a commercially available material that would reduce the magnitude of the tibial shock recorded during the fencing lunge. The results revealed that Surface E (the specifically designed non-absorbent vinyl loop material) was the only material to record significantly reduced tibial accelerations compared to Surface F, (the fencing piste laid atop the concrete convention centre floor). As previous research has suggested the development of

**Table 3. The number of fencers experiencing the greatest and lowest tibial accelerations on each surface as well as the ratio between the two. #denotes the greatest tibial accelerations, *denotes the lowest tibial accelerations.**

| Surface | Greatest (n = 69) | Lowest (n = 69) | Ratio (Greatest to Lowest) |
|---|---|---|---|
| A | 23# | 4 | 5.75 |
| B | 9 | 12 | 0.75 |
| C | 8 | 15 | 0.53 |
| D | 9 | 11 | 0.82 |
| E | 8 | 20* | 0.40 |
| F | 12 | 7 | 1.71 |

overuse injuries is directly related to the frequency and magnitude of impact, it is thus possible that laying Surface E between the fencing piste and the concrete floor of convention centers may potentially reduce injury risk given that it reduced the overall impact shock [10,11].

The exact mechanism by which the non-absorbent vinyl loop material reduces impact shock is unknown; however, it was the least soft material used. When the aluminium fencing piste plate which weighed approximately 11 kg was placed on top of the surfaces it will have compressed them. Surface E, the non-absorbent vinyl loop material will have been compressed the least from the fencing piste plate due to it being firmer, however, the loops will have created air spaces which with a larger force from the lunge could compress. This may have resulted in surface E having the greatest capacity to absorb the accelerations from the lunges compared to the others. This contradicts prior work which has shown that softer materials absorb and dissipate more of the energy from each foot strike, reducing the stress transferred to the body [20]. Impact force is the instantaneous force felt at any given moment during impact. The shorter the window of impact, the greater the impact force. Compressible surfaces lengthen the duration of impact, acting as drag, gradually slowing and removing energy from the colliding body such that the impact force experienced at any given moment is reduced. However, when a material reaches maximum compression, it effectively behaves as a rigid body, no longer able to absorb energy and instead transmitting it. In this case, as the material approaches full compression, the impact is transmitted to the concrete, which, being rigid, returns a reaction force back through the material, abruptly stopping the foot. It is insufficient to assert softer materials to absorb and dissipate more energy from each foot strike. It's important to calibrate material softness for the intended use, as materials that are too soft, provide insufficient resistance and compress too quickly shortening the window of impact and increasing maximum impact force experienced. Conversely, if a material is too firm, it behaves as a rigid and fails to provide adequate cushioning. This begs the more immediate question as to what the optimal level of softness would be to maximally reduce tibial acceleration of the fencing lunge and the more longitudinal question as to whether doing so would, in turn, maximally reduce the number of overuse injuries in the sport.

Results also showed that Surface B (double layer of rubber underlay) had significantly reduced tibial accelerations compared to Surface A (single layer of rubber underlay). Surface B was simply two layers of Surface A, thus suggesting that doubling the thickness of the material under the fencing piste may further reduce tibial accelerations. However, no significant difference was found between Surface D (rebound foam, a double layer of Surface C) and Surface C. While a double layer of Surface E was not explored, future studies should examine multiple layers of the material to better understand the optimal thickness for reducing tibial accelerations of the fencing lunge. These findings align with other previous work investigating how increasing surface thickness can help reduce impact shock. For example, it has been shown that granular flow cushioning found that increasing the thickness of cushioning layers by up to 200 mm could reduce impact shock by 50% [21]. Similar as to understanding the optimal material softness for maximally reducing tibial accelerations of the fencing lunge, the question must also be asked as to what the optimal material thickness is for doing so. However, having a material which is too thick could raise the piste to a height where it could increase risk of injury if the fencer falls

off the edge of the piste, potentially resulting in injury. Future work should thus focus on examining a combination of the optimal material softness and thickness for maximally reducing the impact shock of the fencing lunge while also considering the safety of the height of the piste.

This study is subject to several limitations. Skin tissue artifact/skin resonance associated with skin mounted accelerometry can influence the recording of the underlying bone accelerations [22]. The signal strength measured by the accelerometer is significantly affected by the resonance frequency of its mounting, which complicates interstudy comparisons. Additionally, the axial acceleration is influenced by the centripetal forces caused by tibial angular motion in the sagittal plane during the stance phase [5]. Consequently, even with the device mounted distally, some correction for tibial angular motion may still be needed. Further research is necessary to determine the appropriate adjustments for angular motion during the fencing lunge. Another limitation lies in the use of a 60 Hz Butterworth low-pass filter when processing the acceleration signals. By applying a universal, non-optimized cut-off frequency across all participants, noise is reduced; however, this may also attenuate important high-frequency components of the acceleration signal. The selected cutoff frequency, based on prior studies, represents a compromise between minimizing noise and maintaining signal integrity. High-frequency details, particularly those above 60 Hz, could provide valuable insights into transient forces or tissue resonances. Finally, the shoes worn by the fencers were not standardised, other than wearing shoes they normally fence in (the type and make was recorded), therefore, there were different sole thicknesses which could have affected the impact shock of the lunge.

## Conclusions

The results of this study suggest that overuse injuries caused by repetitive impact shock, particularly in fencing, could potentially be reduced by placing one or more layers of specific materials between the fencing piste and the concrete floors often used at major competitions, such as National USA Fencing tournaments. This finding is especially important for fencers prone to lower body overuse injuries, where frequent, high-impact lunges can result in cumulative stress to the bones and joints. For such athletes, surface modifications could be crucial in prolonging careers and reducing injury downtime. Additionally, these results provide valuable insight for tournament organizers who wish to create safer competition environments, especially in venues with traditionally hard surfaces like concrete. Future investigations should explore how different material properties, including thickness softness and density, interact to achieve the most effective reduction in impact shock. Shoe material properties should also be investigated as it regards to tibial accelerations on multiple surface types to optimize injury prevention and performance in fencing athletes. These investigations will ensure that both safety and performance remain prioritized in the design of fencing competition surfaces.

## Acknowledgments

We would like to thank Kevin Gilmour for 3D printing the tibial shank mounts for the accelerometer. We are grateful to USA Fencing for allowing us to undertake data collection at the National Championships and providing us with space.

## Author contributions

**Conceptualization:** Katharine Holmes, Jonathan Sinclair, Ashley Titan, Lindsay Bottoms.

**Data curation:** Lindsay Bottoms.

**Formal analysis:** Jonathan Sinclair.

**Investigation:** Camilla Holland.

**Methodology:** Katharine Holmes, Jonathan Sinclair, Ashley Titan, Camilla Holland, Michael Grebla, Lindsay Bottoms.

**Project administration:** Katharine Holmes, Ashley Titan, Camilla Holland, Lindsay Bottoms.

**Resources:** Lindsay Bottoms.

**Writing – original draft:** Katharine Holmes, Michael Grebla, Lindsay Bottoms.

**Writing – review & editing:** Katharine Holmes, Jonathan Sinclair, Ashley Titan, Camilla Holland, Michael Grebla, Lindsay Bottoms.

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
