## [Decision Letter · Decision Letter 0]

17 Jan 2025

PONE-D-24-57112The effect of different materials under the fencing piste on impact shock of the tibia during the fencing lunge on a concrete surfacePLOS ONE

Dear Dr. Bottoms,

Thank you for submitting your manuscript to PLOS ONE. After careful consideration, we feel that it has merit but does not fully meet PLOS ONE’s publication criteria as it currently stands. Therefore, we invite you to submit a revised version of the manuscript that addresses the points raised during the review process.

This work explores impact shock experienced by fencers at play. There is is clinical relevance in this manuscript which may raise awareness at methods to mitigate oversuse injuries in this group of athletes. Please see the comments from reviewers below. However, your manuscript has a number of issues which must be addressed before further consideration. Please see reviewer comments below.

We look forward to receiving your revised manuscript.

Kind regards,

Gary Guerra, Ph.D.

Academic Editor

PLOS ONE

3. In the online submission form, you indicated that your data will be submitted to a repository upon acceptance.  We strongly recommend all authors deposit their data before acceptance, as the process can be lengthy and hold up publication timelines. Please note that, though access restrictions are acceptable now, your entire minimal  dataset will need to be made freely accessible if your manuscript is accepted for publication. This policy applies to all data except where public deposition would breach compliance with the protocol approved by your research ethics board. If you are unable to adhere to our open data policy, please kindly revise your statement to explain your reasoning and we will seek the editor's input on an exemption.

4. We note you have included a table to which you do not refer in the text of your manuscript. Please ensure that you refer to Table 2 in your text; if accepted, production will need this reference to link the reader to the Table.

5. We note that Figure 1 includes an image of a [patient / participant / in the study].

Additional Editor Comments (if provided):

Reviewers' comments:

Reviewer's Responses to Questions

**Comments to the Author**

1. Is the manuscript technically sound, and do the data support the conclusions?

Reviewer #1: Yes

Reviewer #2: Partly

2. Has the statistical analysis been performed appropriately and rigorously? 

Reviewer #1: Yes

Reviewer #2: I Don't Know

3. Have the authors made all data underlying the findings in their manuscript fully available?

Reviewer #1: Yes

Reviewer #2: No

4. Is the manuscript presented in an intelligible fashion and written in standard English?

Reviewer #1: Yes

Reviewer #2: Yes

5. Review Comments to the Author

Reviewer #1: 1) Method: Although the filtering method (60 Hz Butterworth low-pass filter) seems to be applied correctly, this methodology should be considered among the potential limitations in the discussion section.

2) Results:

There is a typo in "The results also revelated". It should be "Revealed".

Table 2 should show ratios of numbers rather than percentages; this may provide a more understandable analysis.

3) Discussion: While explaining why Surface E is the least soft material, it was stated that it contradicts the previous findings of "softer materials absorb more energy". This contradiction should be explained by a more detailed mechanism analysis.

4) References:

however, in some sources current data are missing or duplicated (e.g. [7] and [6] appear to have repeated the same study).

5) Spelling errors: There are errors such as "revelated" and "pervious work".

Reviewer #2: This study investigates the effect of various surface materials on tibial accelerations during the fencing lunge, a movement known to contribute to overuse injuries. The experiment involved 69 fencers competing at the 2024 US Senior National Championships, who performed lunges on six different surfaces. The surfaces included combinations of rubber underlay, rebound foam, and non-absorbent vinyl loop coils, placed between the metal fencing piste and concrete floors. The results showed that the non-absorbent vinyl loop material (Surface E) significantly reduced tibial accelerations compared to the standard concrete floor setup, suggesting that modifying competition surfaces with cushioning materials may reduce injury risk.

ABSTRACT

Line 27: Replace parentheses with commas. "This study aimed to investigate tibial accelerations (a measure of impact shock) experienced by.."

Line 30: Replace "performed five lunges on six different surfaces (surfaces A – F; A-E composed of various materials placed between the aluminium piste and the concrete floor; F was only the concrete floor),...." with "performed five lunges on six different surfaces (A–F: A–E composed of various materials placed between the aluminium piste and the concrete floor; F was only the concrete floor)"

INTRODUCTION

Line 49: Which are the four sports to be included? Specify them in the text.

Line 58: add comma "...no padding, and fencers.."

Consider providing more detail or examples of how piste thickness impacts athlete performance or safety.

Line 66: Correct the references 5, 6 and 7.

Line 70: If there is any unpublished article, or one that has only been submitted, it is recommended to cite it.

Lines 77-80: Make the text more cohesive. Try to rephrase it.

METHODS

Lines 91-92: "Seventy fencers (35 female) volunteered to participate in the study (fencers’ characteristics can be seen in Table 1)." Split the two sentences without using parentheses, stating that the anthropometric data are reported in Table 1.

Line 103: In Table 1 caption "Age" should be "age"; "stature" should be "height";

"N" should be "n"

Figure 1: Blur the participant's face.

Include a table with the mechanical properties of the surfaces used (i.e. data sheet with stiffness/hardness, cohesion and friction, dynamic stiffness, dynamic friction factor)

RESULTS

Figure 2: Use square brackets to identify the statistically significant difference between surface A and (B-E) and surface F and E, so that the graph is interpretable without the need to read the figure caption. It could be misinterpreted if the figure caption is not read carefully.

Displaying the pattern of the recorded accelerations could help the reader and facilitate replicating the study for future research.

DISCUSSION

Increase the content of the discussion.

CONCLUSIONS

Include as a possible future study the correlation between the mechanical properties of the shoe material, sole thickness, surface materials, and tibial accelerations.

GENERAL CONSIDERATIONS

Although the lower and higher axial tibial acceleration peaks have been removed, have you considered the effect of different lunge execution speeds on different surface?

Insert the space before the reference. correct: ...word [n]; worng: ...word[n].

Replace the hyphen with the em dash.

It is recommended to use material available in the literature and not rely on data that has been collected but not yet published, especially in the introduction.

6. PLOS authors have the option to publish the peer review history of their article (what does this mean? ). If published, this will include your full peer review and any attached files.

**Do you want your identity to be public for this peer review?** For information about this choice, including consent withdrawal, please see our Privacy Policy .

Reviewer #1: **Yes: ** Esedullah AKARAS

Reviewer #2: No

---

## [Author Response · Author response to Decision Letter 1]

18 Feb 2025

Response to Reviewers

Thank you for the comments to our paper. We have tried our best to make the amendments requested and feel the paper is improved because of it. We outline each comment in this document and how we have responded to it.

Editorial Comments and Responses:

Comment: Please ensure that your manuscript meets PLOS ONE's style requirements, including those for file naming. The PLOS ONE style templates can be found at

Response: I have gone through and tried to make the formatting in PLOS ONE style.

Comment: Please include a complete copy of PLOS’ questionnaire on inclusivity in global research in your revised manuscript. Our policy for research in this area aims to improve transparency in the reporting of research performed outside of researchers’ own country or community. The policy applies to researchers who have travelled to a different country to conduct research, research with Indigenous populations or their lands, and research on cultural artefacts. The questionnaire can also be requested at the journal’s discretion for any other submissions, even if these conditions are not met. Please find more information on the policy and a link to download a blank copy of the questionnaire here: https://journals.plos.org/plosone/s/best-practices-in-research-reporting. Please upload a completed version of your questionnaire as Supporting Information when you resubmit your manuscript.

Response: We have completed this and added as supporting information. We have added Stanford University’s ethics approval as supporting information.

Comment: In the online submission form, you indicated that your data will be submitted to a repository upon acceptance. We strongly recommend all authors deposit their data before acceptance, as the process can be lengthy and hold up publication timelines. Please note that, though access restrictions are acceptable now, your entire minimal dataset will need to be made freely accessible if your manuscript is accepted for publication. This policy applies to all data except where public deposition would breach compliance with the protocol approved by your research ethics board. If you are unable to adhere to our open data policy, please kindly revise your statement to explain your reasoning and we will seek the editor's input on an exemption.

Response: As we have had revisions, I will start this process now.

Comment: We note you have included a table to which you do not refer in the text of your manuscript. Please ensure that you refer to Table 2 in your text; if accepted, production will need this reference to link the reader to the Table.

Response: Apologies, this has been referred to in the paragraph before on page 8.

Comment: We note that Figure 1 includes an image of a [patient / participant / in the study].

Response: As it was one of the authors demonstrating the movement, we thought we wouldn’t need to complete the consent form. Apologies, this has now been completed and added as supplementary material.

Reviewer 1:

Comment: Method: Although the filtering method (60 Hz Butterworth low-pass filter) seems to be applied correctly, this methodology should be considered among the potential limitations in the discussion section.

Response: Thank you for the comment. A limitation on page 10 has been added to highlight this issue.

Comment: There is a typo in "The results also revelated". It should be "Revealed".

Response: Apologies, this has been corrected to revealed.

Comment: Table 2 should show ratios of numbers rather than percentages; this may provide a more understandable analysis.

Response: The ratios have been added as a separate column as we feel it adds to the data rather than removing the total numbers.

Comment: Discussion: While explaining why Surface E is the least soft material, it was stated that it contradicts the previous findings of "softer materials absorb more energy". This contradiction should be explained by a more detailed mechanism analysis.

Response: Thank you for this comment. We have added some text on page 11 to help explain this contradiction. “Impact force is the instantaneous force felt at any given moment during impact. The shorter the window of impact, the greater the impact force. Compressible surfaces lengthen the duration of impact, acting as drag, gradually slowing and removing energy from the colliding body such that the impact force experienced at any given moment is reduced. However, when a material reaches maximum compression, it effectively behaves as a rigid body, no longer able to absorb energy and instead transmitting it. In this case, as the material approaches full compression, the impact is transmitted to the concrete, which, being rigid, returns a reaction force back through the material, abruptly stopping the foot. It is insufficient to assert softer materials to absorb and dissipate more energy from each foot strike. It’s important to calibrate material softness for the intended use, as materials that are too soft, provide insufficient resistance and compress too quickly shortening the window of impact and increasing maximum impact force experienced. Conversely, if a material is too firm, it behaves as a rigid and fails to provide adequate cushioning.”

Comment: References: however, in some sources current data are missing or duplicated (e.g. [7] and [6] appear to have repeated the same study).

Response: Apologies, this should be correct now.

Comment: Spelling errors: There are errors such as "revelated" and "pervious work".

Response: Apologies, these have been corrected and we have proof read the whole article again for spelling mistakes.

Reviewer #2:

Comment: Line 27: Replace parentheses with commas. "This study aimed to investigate tibial accelerations (a measure of impact shock) experienced by.."

Response: Thank you, we have made this amendment to the sentence.

Comment: Line 30: Replace "performed five lunges on six different surfaces (surfaces A – F; A-E composed of various materials placed between the aluminium piste and the concrete floor; F was only the concrete floor),...." with "performed five lunges on six different surfaces (A–F: A–E composed of various materials placed between the aluminium piste and the concrete floor; F was only the concrete floor)"

Response: Thanks for the amendment, this has been made.

Comment: Line 49: Which are the four sports to be included? Specify them in the text.

Response: This has been added to the text.

Comment: Line 58: add comma "...no padding, and fencers.."

Response: Added, thank you.

Comment: Consider providing more detail or examples of how piste thickness impacts athlete performance or safety.

Response: Thank you for the thought-provoking comment. However, there is no research which shows the thickness of the piste affects athlete safety. We only have data from a previous study which is the one discussed in the intro on page 4.

Comment: Line 66: Correct the references 5, 6 and 7.

Response: There was a mistake here with the reference manager used, this has been corrected.

Comment: Line 70: If there is any unpublished article, or one that has only been submitted, it is recommended to cite it.

Response: We have replaced the unpublished research with an older paper which has similar findings.

Comment: Lines 77-80: Make the text more cohesive. Try to rephrase it.

Response: This has been rewritten to make it more cohesive.

Comment: Lines 91-92: "Seventy fencers (35 female) volunteered to participate in the study (fencers’ characteristics can be seen in Table 1)." Split the two sentences without using parentheses, stating that the anthropometric data are reported in Table 1.

Response: This has been amended in the text and is now 2 sentences.

Comment: Line 103: In Table 1 caption "Age" should be "age"; "stature" should be "height";

"N" should be "n"

Response: Apologies, this has been amended.

Comment: Figure 1: Blur the participant's face.

Response: This is a photo of one of the authors and we have now completed a consent form for the photo and therefore do not need to blur the photo now.

Comment: Include a table with the mechanical properties of the surfaces used (i.e. data sheet with stiffness/hardness, cohesion and friction, dynamic stiffness, dynamic friction factor)

Response: We have changed the text to a table and added as many characteristics as we have available to us.

Comment: Figure 2: Use square brackets to identify the statistically significant difference between surface A and (B-E) and surface F and E, so that the graph is interpretable without the need to read the figure caption. It could be misinterpreted if the figure caption is not read carefully.

Displaying the pattern of the recorded accelerations could help the reader and facilitate replicating the study for future research.

Response: We have added the brackets to Figure 2 which hopefully makes it more reader friendly. However, we can’t see how we can add the pattern of the recorded accelerations. Hopefully, the figure as is does the job.

Comment: Increase the content of the discussion.

Response: With the added text from the other comments we feel this has increased the content of the discussion. We feel the discussion addresses the explanation of the results and the potential impact.

Comment: CONCLUSIONS: Include as a possible future study the correlation between the mechanical properties of the shoe material, sole thickness, surface materials, and tibial accelerations.

Response: That’s an excellent suggestion. Including a future study on the correlation between the mechanical properties of shoe materials, sole thickness, surface materials, and tibial accelerations could provide critical insights into injury prevention and performance optimization. Unfortunately, we did not collect shoe type information for this study.

Understanding how these factors interact could help identify the combinations that minimize tibial accelerations, thereby reducing the risk of stress fractures and other overuse injuries. This research could also inform the design of sport-specific footwear and surfaces to enhance safety and comfort for fencing athletes.

In the text we now include the following: “Shoe material properties should also be investigated as it regards to tibial accelerations on multiple surface types to optimize injury prevention and performance in fencing athletes.”

Comment: Insert the space before the reference. correct: ...word [n]; worng: ...word[n].

Replace the hyphen with the em dash.

Response: This has been amended throughout.

---

## [Decision Letter · Decision Letter 1]

2 Mar 2025

PONE-D-24-57112R1The effect of different materials under the fencing piste on impact shock of the tibia during the fencing lunge on a concrete surfacePLOS ONE

Dear Dr. Bottoms,

Thank you for submitting your manuscript to PLOS ONE. After careful consideration, we feel that it has merit but does not fully meet PLOS ONE’s publication criteria as it currently stands. Therefore, we invite you to submit a revised version of the manuscript that addresses the points raised during the review process.

We look forward to receiving your revised manuscript.

Kind regards,

Gary Guerra, Ph.D.

Academic Editor

PLOS ONE

Journal Requirements:

Reviewers' comments:

Reviewer's Responses to Questions

**Comments to the Author**

1. If the authors have adequately addressed your comments raised in a previous round of review and you feel that this manuscript is now acceptable for publication, you may indicate that here to bypass the “Comments to the Author” section, enter your conflict of interest statement in the “Confidential to Editor” section, and submit your "Accept" recommendation.

Reviewer #1: All comments have been addressed

Reviewer #2: All comments have been addressed

2. Is the manuscript technically sound, and do the data support the conclusions?

Reviewer #1: Yes

Reviewer #2: Partly

3. Has the statistical analysis been performed appropriately and rigorously? 

Reviewer #1: Yes

Reviewer #2: N/A

4. Have the authors made all data underlying the findings in their manuscript fully available?

Reviewer #1: Yes

Reviewer #2: No

5. Is the manuscript presented in an intelligible fashion and written in standard English?

Reviewer #1: Yes

Reviewer #2: Yes

6. Review Comments to the Author

Reviewer #1: (No Response)

Reviewer #2: Adding a figure displaying the pattern (mean and SD as a function of time) of the recorded accelerations (not only the peaks, as in Figure 2) could help the reader and facilitate the replication of the study in future research.

Even if an author is present in the photo, it is advisable to blur their face to ensure anonymity and respect privacy.

7. PLOS authors have the option to publish the peer review history of their article (what does this mean? ). If published, this will include your full peer review and any attached files.

**Do you want your identity to be public for this peer review?** For information about this choice, including consent withdrawal, please see our Privacy Policy .

Reviewer #1: **Yes: ** Esedullah AKARAS

Reviewer #2: No

---

## [Author Response · Author response to Decision Letter 2]

6 Mar 2025

Response to Reviewers

Thank you for the comments to our paper.

Editorial Comments and Responses:

Response: We have not made any amendments and the references seem to be correct.

Reviewer comments:

Reviewer #2:

Reviewer: Adding a figure displaying the pattern (mean and SD as a function of time) of the recorded accelerations (not only the peaks, as in Figure 2) could help the reader and facilitate the replication of the study in future research.

Response: We have discussed this as a team but we do not feel it appropriate to add this figure. When we wrote the ethics application we planned to analyse the peak impact accelerations as this is what we are most interested in. We therefore do not want to deviate away from our pre planned analyses. Secondly, this is the method we have previously used and published (Sinclair et al., 2010 and Greenhalgh et al., 2013). Thirdly, to do this figure would mean rerunning all the analyses and using SPM method for statistics. Also, we do not have the exact start and end times for the lunges as we have done this in the field which means we can’t accurately time normalise it. Therefore, this wouldn’t provide any additional clarity to the reader. We feel our current figure is the most appropriate to best represent the data.

Reviewer: Even if an author is present in the photo, it is advisable to blur their face to ensure anonymity and respect privacy.

Response: We have now blurred the face of the fencer.

---

## [Decision Letter · Decision Letter 2]

10 Apr 2025

The effect of different materials under the fencing piste on impact shock of the tibia during the fencing lunge on a concrete surface

PONE-D-24-57112R2

Dear Dr. Bottoms,

We’re pleased to inform you that your manuscript has been judged scientifically suitable for publication and will be formally accepted for publication once it meets all outstanding technical requirements.

Kind regards,

Gary Guerra, Ph.D.

Academic Editor

PLOS ONE

Additional Editor Comments (optional):

Reviewers' comments:

Reviewer's Responses to Questions

**Comments to the Author**

1. If the authors have adequately addressed your comments raised in a previous round of review and you feel that this manuscript is now acceptable for publication, you may indicate that here to bypass the “Comments to the Author” section, enter your conflict of interest statement in the “Confidential to Editor” section, and submit your "Accept" recommendation.

Reviewer #1: All comments have been addressed

Reviewer #3: All comments have been addressed

2. Is the manuscript technically sound, and do the data support the conclusions?

Reviewer #1: Yes

Reviewer #3: Yes

3. Has the statistical analysis been performed appropriately and rigorously? 

Reviewer #1: Yes

Reviewer #3: Yes

4. Have the authors made all data underlying the findings in their manuscript fully available?

Reviewer #1: Yes

Reviewer #3: Yes

5. Is the manuscript presented in an intelligible fashion and written in standard English?

Reviewer #1: Yes

Reviewer #3: Yes

6. Review Comments to the Author

Reviewer #1: (No Response)

Reviewer #3: This manuscript has been thoroughly revised and appears to have addressed all the major concerns and suggestions raised by the reviewers during the previous round of review. The authors have responded appropriately and made the necessary modifications, resulting in a significantly improved and more robust version of the manuscript.

7. PLOS authors have the option to publish the peer review history of their article (what does this mean? ). If published, this will include your full peer review and any attached files.

**Do you want your identity to be public for this peer review?** For information about this choice, including consent withdrawal, please see our Privacy Policy .

Reviewer #1: **Yes: ** Esedullah AKARAS

Reviewer #3: No

---

## [Editor Report · Acceptance letter]

PONE-D-24-57112R2

PLOS ONE

Dear Dr. Bottoms,

I'm pleased to inform you that your manuscript has been deemed suitable for publication in PLOS ONE. Congratulations! Your manuscript is now being handed over to our production team.

Kind regards,

on behalf of

Dr. Gary Guerra

Academic Editor

PLOS ONE